# Natural Bioactives: Back to the Future in the Fight against Human Papillomavirus? A Narrative Review

**DOI:** 10.3390/jcm11051465

**Published:** 2022-03-07

**Authors:** Silvia Massa, Riccardo Pagliarello, Francesca Paolini, Aldo Venuti

**Affiliations:** 1Biotechnology Laboratory, Casaccia Research Center, Biotechnology and Agro-Industry Division, Italian National Agency for New Technologies, Energy and Sustainable Economic Development (ENEA), 00123 Rome, Italy; riccardo.pagliarello@enea.it; 2Department of Agriculture and Forest Sciences (DAFNE), University of Tuscia, 01100 Viterbo, Italy; 3HPV-Unit, Unità Operativa Semplice Dipartimentale (UOSD) Tumor Immunology and Immunotherapy, IRCCS Regina Elena National Cancer Institute, 00144 Rome, Italy; francesca.paolini@ifo.gov.it (F.P.); aldo.venuti@ifo.gov.it (A.V.)

**Keywords:** HPV, natural, marine, algal, microbial compounds, phytochemicals, bioactives, chemoprevention, combination treatments

## Abstract

Human papillomavirus (HPV) still represents an important threat to health worldwide. Better therapy in terms of further improvement of outcomes and attenuation of related side-effects is desirable. The pharmaceutical industry has always targeted natural substances—phytochemicals in particular—to identify lead compounds to be clinically validated and industrially produced as antiviral and anticancer drugs. In the field of HPV, numerous naturally occurring bioactives and dietary phytochemicals have been investigated as potentially valuable in vitro and in vivo. Interference with several pathways and improvement of the efficacy of chemotherapeutic agents have been demonstrated. Notably, some clinical trials have been conducted. Despite being endowed with general safety, these natural substances are in urgent need of further assessment to foresee their clinical exploitation. This review summarizes the basic research efforts conducted so far in the study of anti-HPV properties of bio-actives with insights into their mechanisms of action and highlights the variety of their natural origin in order to provide comprehensive mapping throughout the different sources. The clinical studies available are reported, as well, to highlight the need of uniformity and consistency of studies in the future to select those natural compounds that may be suited to clinical application.

## 1. Introduction

Cancer is the main cause of death in wealthy countries with a constant increase in the number of cases around the globe. In 2020, 19.3 million new cancer cases were estimated (18.1 million, omitting nonmelanoma skin cancer) and almost 10 million cancer deaths (9.9 million, omitting nonmelanoma skin cancer) occurred [1]. About 15–20% of all human cancers are related to viral infections [2,3] and human papillomavirus (HPV)-related cancers account for 5% of all human cancers, with HPV being the most common sexually transmitted virus [4]. HPV tumors mainly originate from persistent infection with high-risk types HPV-16 and HPV-18 [5,6]. HPV infection is linked to anogenital cancers (cervical, vulvar, vaginal, anal, and penial) and head and neck cancers in both men and women [4,7]. Among HPV-associated tumors, cervical cancer still ranks fifth in terms of both incidence (6.5% of total female cases) and mortality (7.7% of total female deaths for cancer) of all cancers in the female population [7]. Information about transmission, cytological screening, and preventive vaccination are essential factors in limiting HPV infection and tumor spread in the global population [8], and thus in reducing the related incidence and mortality [9].

Despite preventive vaccination having been a big step in the direction of HPV cancer prevention [10], the incidence of HPV-related cancers continues to rise. Thus, HPV disease is still considered a serious public health concern [2]. Suggested for women up to 45 years of age, but mainly administered in the 9–15 years cohort, the impact of vaccination is visible mainly in the long run. The options available for therapy of HPV cancers (i.e., surgery, and chemo- and radio-therapy) are insufficient. Novel promising targeted agents with potential anti-HPV cancer effects comprise repurposed drugs (i.e., rimantadine) [11], in silico defined small molecules [12], and immunotherapeutics such as CAR-T cells, immune checkpoint inhibitors (ICIs), single-chain intrabodies, and therapeutic vaccines [13,14,15,16].

Taking advantage of the presence of a reliable tumor-associated antigen, represented by HPV E6 and E7 oncoproteins, the development of therapeutic vaccines able to provide better or comparable results to those treatments already in use which would be highly feasible and desirable. Many experimental efforts to develop therapeutic HPV vaccines have shown promising results, but no therapeutic vaccines are currently licensed for use [17]. Among therapeutic experimental HPV vaccines, ‘plant-inspired’ formulations have been developed, demonstrating the promise of plant-based natural compounds in making immunity more oriented to mount a response to the vaccine. As an example, a *Nicotiana benthamiana* crude extract containing HPV-16 E7 protein induced a cell-mediated immune response, protecting mice from experimental tumor challenge, in absence of concomitant adjuvant administration [18,19]. The same extract induced the maturation of human dendritic cells that were then able to induce human blood-derived T lymphocytes from healthy individuals to exert HPV-16 E7-specific cytotoxic activity [20]. A similar increased immune “visibility” was proven for an E7 protein-based vaccine produced through the use of the *Chlamydomonas*
*reinhardtii* microalga [21]. The common feature of these plant-based vaccines is administration in the form of extracts and their intrinsic adjuvant activity [22].

Increasing attention is being given to the possible impact of purified natural substances and extracts on the development of HPV tumors [23,24,25]. The prophylactic inhibitory or reverting effect exerted by synthetic, natural, or biological chemicals against carcinogenesis commonly defines the chemoprevention ability of a substance. Chemoprevention may have importance in reducing, preventing, and inhibiting the development of different types of cancer, as nearly 33% of them can be avoided by lifestyle changes, including diet [26]. The ideal chemopreventive compound should be safe, highly efficient, and available at low production cost. Oral administration with diet would be a useful ‘plus’. Very often, indeed, the heavier budget items in the production of plant-based compounds include purification and downstream processing. This is the reason why population studies on plasma levels of phytochemicals and natural diet-related substances have been carried out in this field as well.

In addition to decreasing the risk of cancer development, natural compounds can influence one given routine chemo- and radio-therapeutic treatment by improving the sensitivity of tumors to chemo- and/or radio-therapeutics and, as a consequence, enhance related beneficial effects [27,28]. These effects are known as chemo- and radio-sensitization.

It is worth noting that many phytochemicals have antioxidant activities that, by eliminating free oxygen radicals, prevent cell DNA damage and, therefore, counteract one of the mechanisms of carcinogenesis [29]. Hence, antioxidant phytochemicals, beside reducing the risk of cancer development per se, can improve the cytotoxic effects of chemotherapeutics [22,23].

Despite research on plant-derived natural substances having been documented for years in the field of HPV, the real clinical applications of these substances still need further research and organic comprehension (for a list and details of published clinical trials grouped by formulation, see Appendix A). The current lack of longitudinal results, especially from in vivo and clinical points of view, does not allow us to foresee immediate implications for clinical practice. As a consequence, this review aims to highlight the efforts done in the study of anti-HPV properties of bioactives from different sources, and to provide a more comprehensive mapping of these properties. In fact, besides the well-known compounds of plant origin, the potential application of bacterial-derived and marine-bioactive compounds has been shown against HPV as well (Figure 1). In addition, the anti-HPV potential of some natural compounds has also been shown in silico by molecular modelling.

This review also offers a spotlight on plant (but also microbial) cells as tools to set up biotechnological factories of value-added specialized metabolites and proteins when naturally occurring sources are rare, endangered, only seasonally available, or when these molecules need qualitative or quantitative improvement.

## 2. Methodology

### 2.1. Criteria for Reference Selection

The search for reference material was carried out by consulting online databases for existing work relevant to the topic. PubMed, MEDLINE, ScienceDirect, Scopus, and Google Scholar (the latter in order to identify both published and unpublished information) were searched, applying publication date and other filters. In order to obtain a broader range search, also older publications were considered, when relevant. To minimize the risk of bias that may be associated with narrative reviews, our search was comprehensive in order to identify both research studies as well as reviews. This search was performed through an analysis of keywords throughout the specific index terms related to the articles and the words contained in the whole text of the retrieved papers.

### 2.2. Research Question

Search query was performed using the following keywords: HPV disease, natural compounds, bioactives, phytochemicals, combination treatments, therapy, infection, name of compounds, and class of compounds.

### 2.3. Study Selection

The results of the database searches were transferred to a Mendeley library and then were de-duplicated. Irrelevant records were removed before identifying records that met the inclusion criteria. Inclusion criteria involved reviews and/or research papers on in vitro/in vivo/clinical randomized and non-randomized studies published in indexed and peer-reviewed scientific journals. Exclusion criteria involved abstracts or unpublished works with insufficient or invalid information established on the basis of the expertise of co-authors.

### 2.4. Charting the Data

Data were synthesized and interpreted using a narrative/descriptive approach. All published papers matching the above-mentioned criteria have been quoted in the manuscript and in the bibliography section, after a critical revision of the exposure experimental methods.

## 3. Results

Results obtained from literature scan following the Methods described in Section 2 are provided in Table 1. 

Results were summarized using an analytic approach to guide the narrative account of existing literature. Due to the huge availability of entities tested against HPV, for ease of consultation, compounds and formulations tested in the different studies have been reported in alphabetical order by chemical family and type of formulation (i.e., purified entities alone or in combination, extracts, mixtures), describing natural source, type of study through which they have been investigated (i.e., in vitro, in vivo, case study, clinical trial) and related outcomes. For completeness and ease of consultation, a summary of all the phytocompounds (i.e., the vast majority of natural chemical entities tested against HPV) reported in the present review, together with further phytocompounds cited by additional literature, is reported in Appendix A.

## 4. Purified Phytochemicals

Since HPV-18-positive HeLa cells were discovered, and before the HPV etiology was established, phytochemicals and traditional ethnobotanical compounds have been tested against cervical cancer [30,31,32]

Plant cells have the highest diversity of metabolite production among live cells [33,34]. Indeed, plants necessitate specialized metabolites for their development and survival. Phytochemicals, as purified and characterized entities (i.e., the so-called specialized metabolites) (Figure 2), will be reviewed in this section (for a complete list, see Appendix A).

### 4.1. Polyphenols

Numerous in vivo and/or in vitro studies have mainly focused on the anticancer effects of purified plant-derived metabolites and, specially, on polyphenols, most of which are flavonoids [23,24,25]. This group of phenolic compounds, which is easily encountered in daily nutrition, is diverse and is composed of sub-groups of compounds. Polyphenols will be presented in their outcomes through listing by their four major classes, starting from the initial biosynthetic steps of the pathway: phenolic acids, flavonoids, stilbenes, lignans, and curcuminoids. The effects of these compounds have been demonstrated to rely on the activation or inhibition of several pathways and, in some cases, their action has been demonstrated to be exerted directly on E6 and E7 oncoproteins or their targets.

#### 4.1.1. Phenolic Acids

**Caffeic acid** (hydroxycinnamic acid) from *Ocimum gratissimum* (1 mM and 10 mM), has been demonstrated to significantly reduce the proliferation of HeLa cells by apoptosis through decreased levels of caspase-3 and Bcl-2, and upregulation of caspase-3 and p53 in a concentration-dependent manner, triggered 24–48 h after exposure. This indicates that caffeic acid induces apoptosis via the mitochondrial apoptotic pathway [35].

Caffeic acid has shown induction of cervical cancer cell death in vitro by generating reactive oxygen species (ROS), arresting cycle in S and G2/M phases, and by binding to histone deacetylase (HDAC) followed by apoptosis induction, as well [36].

**Ellagic acid** is found in strawberries, raspberries, walnuts, and pomegranate. In particular, ellagic acid from pomegranate peel has been shown to induce inhibition of HeLa cells by blocking the AKT/mTOR signaling pathway through enhanced expression of insulin-like growth factor-binding protein 7 (IGFBP7). The invasion ability and apoptosis rate of treated HeLa cells was demonstrated, indeed, to be significantly lower compared to controls [37]. However, the authors reported that the ellagic acid use has some limitations due to intrinsic characteristics such as poor absorption and bioavailability and lack of information regarding its safety in the long term.

**Ferulic acid** (hydroxycinnamic acid) has been shown to significantly inhibit proliferation and invasion of HeLa and CaSki cells through the reduction in MMP-9 mRNA expression and by an induced arrest in G0/G1 phase, also inhibiting autophagy in both cancer cell types in a dose-dependent mechanism (*p* < 0.05) [38].

In addition, ferulic acid has been demonstrated to increase the radiosensitivity of HeLa and Me180 cells by potentiation of apoptosis through a ROS-dependent mechanism [39].

**Gallic acid** is a hydroxybenzoic acid found in grapes, blackberries, raspberries, pomegranate, tea, vanilla, and in the leaves of many plants. This compound has demonstrated induction of apoptosis and necrosis in HeLa cells accompanied by ROS increase and glutathione (GSH) depletion [40].

Gallic acid-related downregulation of epidermal growth factor (EGFR), phosphorylated serine/threonine kinase 1 (Akt/p-Akt), and phosphorylated extracellular signal-regulated kinase (Erk/p-Erk) signaling pathways has been proposed as a possible tool to induce suppression of cancer progression in vivo as well [41].

#### 4.1.2. Flavonoids

##### Flavanones

**Hesperetin** is mainly found in *Citrus* species and has been demonstrated to induce apoptosis in SiHa cells by upregulating caspases, p53, Bax, and Fas death receptor and its adaptor protein [42,43].

**Naringenin**, mainly accumulated in grapefruit and citrus fruits, has been shown to inhibit the proliferation of SiHa cells by cell cycle arrest at the G2/M phase and apoptosis triggering [44]. Due to its intrinsic instability, studies have been conducted on naringenin-loaded nanoparticles. The use of these nanoparticles has shown better outcomes over free naringenin in HeLa cells [45] where they were able to determine dose-dependent cytotoxicity, apoptosis, decreased intracellular glutathione levels, alterations in mitochondrial membrane potential, increased intracellular levels of ROS, and lipid peroxidation. Naringenin was found to induce apoptosis of HeLa cells as well by reducing expression of NF-κB p65 subunit, COX-2, and caspase-1. In addition, naringenin treatment induced p21WAF1 expression and downregulated expression of cyclins and cyclin-dependent kinases (CDKs). Moreover, treatment with naringenin induced phosphorylation of extracellular signal-regulated kinase (ERK), p38 mitogen-activated protein kinase, and c-Jun N-terminal kinase. Despite these results, naringenin application in cancer is limited due to its suboptimal pharmacokinetics and poor bioavailability at the tumor sites [46,47,48,49].

##### Flavones

**Apigenin**, the most active flavone in plants, is found in fruits and vegetables and it is of particular interest due to its lower intrinsic toxicity [50,51,52,53,54]. It has been extensively tested in vitro.

CaSki, HeLa, and C-33A growth was inhibited by G1 phase arrest, p53-dependent apoptosis, and upregulation of p21WAF1 expression, Fas/APO-1 death receptor, and caspase-3. Apigenin has been demonstrated to downregulate the antiapoptotic factor Bcl-2 [55].

Interference with HeLa cell gap junctions has been considered responsible for the apigenin-induced attenuation of their motility and translocation in vitro, revealing a possible tool to reduce invasive potential in vivo through this phytocompound [56]. Apigenin has also been demonstrated to sensitize HeLa and SiHa cells to paclitaxel through apoptosis induction triggered by intracellular ROS accumulation [57]. Upon apigenin administration, TC-1 tumor cells (primary mice lung cells transformed with HPV-16 E6, HPV-16 E7, and ras oncogenes) have been demonstrated to become more susceptible to lysis by E7-specific cytotoxic CD8+ T cells and to apoptosis in vitro in a dose-dependent fashion.

In the context of a combinatorial in vivo pre-clinical study, administration of apigenin in association with an E7-HSP70 DNA vaccine has been shown to increase the levels of primary and memory E7-specific CD8+ T cells, leading to therapeutic anti-tumor effects against E7-expressing tumors in TC-1 tumor-bearing mice [58].

**Jaceosidin**, common in *Compositae*, has been shown to block E6 and E7 oncoproteins activity by impeding binding to p53 and pRb in SiHa and CaSki cells [59]. In addition, jaceosidin increased cleavage of caspase-9 and caspase-3 in oral squamous carcinoma cells and downregulated Akt phosphorylation. Jaceosidin demonstrated no effects on HaCaT normal epithelial cell viability, indicating a possible selective chemotherapeutic potential [60].

**Luteolin** is most often found in perilla, broccoli, celery, parsley, thyme, olive oil, carrots, oregano, oranges, and peppermint. It has been demonstrated to induce apoptosis in HeLa cells and to be an E6 inhibitor through binding to a hydrophobic pocket at the interface between E6 and E6AP [61]. In particular, luteolin enhanced the expression of death receptor downstream factors—such as Fas/FasL, DR5/TRAIL, and FADD in HeLa cells—and activated caspase cascades, especially caspase-3 and caspase-8 in a dose-dependent manner.

**Wogonin** from *Scutellaria baicalensis* has been found to trigger apoptosis through suppression of E6 and E7 and increased p53 and pRb in SiHa and CaSki cells [62]. Wogonin demonstrated A549 and HeLa cell sensitization to cisplatin by ROS generation as well [63].

##### Flavonols

**Epigallocatechin-3-gallate (EGCG)** is considered the most bioactive agent of green tea from *Camellia sinensis*, and it has shown antiproliferative, antiangiogenic, anti-metastatic, and proapoptotic effects by affecting a wide array of signal transduction pathways including JAK/STAT, MAPK, PI3K/AKT, Wnt, and Notch [64]. The inhibition of HPV E6/E7 expression and estrogen receptor α and aromatase is supposed to be the main mechanism [65] of EGCG-mediated apoptosis in HPV cancer cells [66]. It has been demonstrated that the cell type involved in cervical cancer development can affect the result of the treatment with this compound. Indeed, when treated with 50 µg/mL of EGCG, cell growth was inhibited by 100% in squamous cell carcinoma cell line Me180, while growth inhibition was less evident in adenocarcinoma cell line HeLa (85.6%) [67]. A genome-wide study has demonstrated that EGCG also affected DNA methylation in the oral squamous cell carcinoma CAL 27 cell line [68]. EGCG was able to suppress HeLa, CaSki, and C-33A cells by regulating the expression of miRNAs [69].

A combination of tea polyphenols with EGCG sensitized HeLa cells to bleomycin by enhanced apoptosis induction triggered through caspase-3, -8, -9, and by upregulation of p53 and Bcl-2 [70].

In addition, a study associating EGCG with E7-based DNA vaccination demonstrated enhanced tumor-specific T-cell response and antitumor effects relative to either DNA vaccine or EGCG alone in tumor-bearing mice. Combined DNA vaccination and oral EGCG treatment provided long-term antitumor protection in healed mice that were afterward able to specifically reject experimental E7-expressing tumors, showing antigen-specific immune responses [71].

**Fisetin**, found mainly in strawberries, apples, onions, cucumbers, and khaki, has shown HeLa anti-proliferation properties through apoptosis mediated by inhibition of ERK1/2 and activation of caspase-8 and-3 [72]. Moreover, treatment of HeLa cells with fisetin induced a sustained activation of the phosphorylation of ERK1/2, and inhibition of ERK1/2 by PD98059 (MEK1/2 inhibitor).

**Kaempferol**, found in many fruits and vegetables, has been shown to block HeLa cell proliferation by G2/M phase cell cycle block, decreasing cyclin B1 and CDK1, inhibition of NF-kB nuclear translocation, upregulation of Bax, and downregulation of Bcl-2 [73]. In addition, kaempferol inhibited the proliferation of SiHa cells in a time- and dose-dependent manner—inducing cell apoptosis, mitochondrial membrane potential disruption, and intracellular free calcium elevation [74].

**Quercetin**, which is found in many fruits, vegetables, and grains, has been demonstrated to induce apoptosis through G2/M phase block, inhibition of anti-apoptotic AKT, and Bcl-2 expression in HeLa cells [75]. Quercetin sensitized HeLa cells to apoptosis upon paclitaxel treatment by an enhancement of intracellular ROS level [76]. Quercetin has also been shown to increase the radiosensitivity of HeLa cells in vitro and of DLD1 human colorectal cancer xenografts in vivo by enhancing apoptosis through a time- and dose-dependent mechanism determining ROS modulation and downregulation of E6 and E7 expression [77].

##### Isoflavonoids

**Daidzein** is represented in legumes (mainly in soybeans) and is the most frequent isoflavone in nature. At 6.25–100 µmol/L concentrations, it blocked the growth of HeLa cells affecting the cell cycle and downregulation of telomerase catalytic subunit mRNA, especially at the lowest concentrations [78].

**Formononetin**, accumulated by several plants and herbs such as red clover, has been demonstrated to sensitize cervical cancer (HeLa) cells in vitro to epirubicin via ROS production [79].

**Genistein**, mostly found in legumes, has been shown to block the growth of CaSki and HeLa cells in vitro through cell cycle arrest and activation of the AKT gene [80,81,82]. However, the safety of genistein demonstrated in this work was judged controversial as the concentration tested was higher than that reported to have anti-cancer effects (i.e., as tested against MCF-7 cells) in literature. The effect of genistein on cancer cells is still controversial, since a study reported a stimulatory effect of genistein and quercetin on MCF-7 cells [83,84].

##### Anthocyanins

These natural pigments are mainly found in black currant, eggplant, cherry, cranberries, blueberries, and black rice and are known for their ROS scavenging ability. In the case of HPV, cyanidin 3-glucoside from black rice has been demonstrated to inhibit proliferation of HeLa cells in a dose- and time-dependent manner by apoptosis induced by Bax/Bcl-2 [85].

##### Flavolignans

**Silibinin** has shown cell cycle modulation (G2 arrest) and dose- and time-dependent apoptosis induction abilities via the activation of the dynamin-related protein 1 in HeLa cells [86,87].

##### Lignans

**Methylenedioxy lignan** showed inhibition of HeLa cell proliferation through apoptosis and inhibition of telomerase [88]. In addition, methylenedioxy lignan inhibits Bcl2 and activates caspase-3 and caspase-8.

**Nor-dihydro-guaiaretic acid** from *Larrea tridentata* downregulated HPV E6 and E7 expression and promoted apoptosis in SiHa cells through suppression of HPV transcription. In addition, nor-dihydro-guaiaretic acid induced cell cycle arrest at the G1 phase and promoted acetylation of histone H3 and p21 gene-associated chromatin [89].

**Sesamin**, a major lignan from sesame oil, has been demonstrated to inhibit the proliferation and migration of HeLa cells and to induce ER stress-mediated apoptosis through the IRE1alpha/JNK pathway. Furthermore, it was able to activate autophagy [90].

#### 4.1.3. Stilbenes

**Resveratrol** accumulates in grapes, blueberries, raspberries, and peanuts. In HeLa cells, it reduced the generation of ROS and suppressed invasion and migration [91]. In CaSki cervical cancer cells it induced a decreased expression of metalloproteinases. Inhibition of AKT and ERK1/2, decrease in the angiogenic activity, destabilization of lysosomes, induction of autophagy, and increased cytosol translocation are other activities that resveratrol was found to exert in the above-mentioned cervical cancer cells [92]. Resveratrol has shown increased radiosensitivity and potentiation of apoptosis of HeLa and SiHa cells through dose-dependent increased cytotoxicity and cell cycle alteration [93]. However, doubts have arisen regarding the characteristics of the patients enrolled, the dose used and duration of supplementation in the above-mentioned study, as well as resveratrol toxicity at high doses [94].

#### 4.1.4. Curcuminoids

**Curcumin** is a hydrophobic polyphenol derived from the rhizome of *Curcuma longa*. It shows several pharmacological properties.

The effect of curcumin on tumor HeLa, SiHa, and C-33A cells has been found to involve the induction of apoptosis and the downregulation of COX-2. HPV-specific effects have been demonstrated for HeLa and SiHa cells that were able to downregulate the viral oncogenes E6 and E7, to inhibit TNF-α-induced NFκB and EGF-induced AP-1 activation, iNOS, and cyclin D1 [95].

Curcumin also exerts downregulation of HPV-18 transcription, inhibition of AP-1 binding activity and expression of c-fos and fra-1 in HeLa cells [96].

In addition, curcumin has been found to upregulate Bax, promote the release of cytochrome c, and downregulate Bcl-2 and Bcl-XL, inducing cell growth inhibition and apoptosis [97]. The cytotoxic potential of curcumin analogs such as diarylpentanoids has been studied in HeLa and CaSki cells as well. Among them, 1,5-Bis(2-hydroxyphenyl)-1,4-pentadien-3-one has been demonstrated to have cytotoxic, anti-proliferative, and apoptotic effects in CaSki cells due to activation of caspase-3. Downregulation of HPV-18 and HPV-16 E6 and E7 oncogene expression has been found as well [98].

Curcumin has been demonstrated to be a sensitizing agent to taxol- and to cisplatin for cervical cancer (HeLa) cells by augmenting apoptosis through downregulation of NF-kB [99,100,101]. Tetrahydrocurcuminoids increased the sensitivity to vinblastine, mitoxantrone, and etoposide of a drug-resistant human cervical carcinoma cell line [102]. Curcumin has also been demonstrated to increase radiosensitivity and apoptosis in HeLa and SiHa cells through an ROS-dependent mechanism [103].

Highly purified curcumin has been also evaluated in clinical setting in a phase I clinical trial of oral administration of 0.5–12 mg for 3 months in four cervical intraepithelial neoplasia (CIN) cases, as well. This dosage has been demonstrated to be clinically safe and has resulted in histological improvements in one out of four cases. Bioavailability and toxicology were also investigated, concluding that the safe dose of curcumin may be up to 8 mg by day (Appendix A, curcumin-based formulation no.1) [104].

### 4.2. Other Specialized Metabolites

Although polyphenols may hold the crown of the most tested compounds against HPV in vitro, many other phytochemical species have been investigated, amplifying the panorama of tools provided by plant compounds in this field.

#### 4.2.1. Terpenoids

Several case-control studies have shown significant negative correlation between consumption of terpenoids such as carotenoids (naturally found in sweet potatoes, carrots, dark leafy greens, lettuce, red peppers, tomato, etc.), retinol (accumulating in amaranth, spinach and chard, orange-fleshed sweet potatoes, carrots, pumpkins, yellow maize, mango, and papaya), and tocopherols (retrievable in many plant-derived oils, nuts, seeds, fruits, and greens) antioxidant micronutrients and risk of cervical cancer [105,106,107,108].

**Beta-carotene**, tocopherols, and retinol have shown no association with risk of cervical cancer from a prospective dietary study [109].

Since studies implying dietary administration of compounds may suffer from inaccuracy in measuring their effective intake and bioavailability, researchers have proposed to put in relationship effective, constitutive plasma levels of recruited patients with the risk of cervical cancer.

**Alpha-carotene**, **β-carotene**, **lycopene**, **or tocopherols** have shown an inverse association between serum levels and cervical cancer risk in two studies [108,110]. The same findings had not been obtained by Lehtinen and colleagues [111], probably due to the not very consistent construction of the study.

**Alpha-carotene, beta-carotene, lutein, zeaxanthin, and alpha-tocopherol** plasma levels, individually, have been evaluated in a study enrolling Chinese women. Constitutive plasma levels of these compounds in patients with histologically confirmed cervical cancer before the initiation of radiotherapy or chemotherapy were compared with plasma levels of these anti-oxidants in healthy controls with no history of cervical cancer. A statistically significant inverse correlation was found between plasma levels of these compounds and diagnosis of cervical cancer in a case-control study. In particular, alpha-carotene (odds ratios = 0.50, 95%; confidence interval = 0.38–0.67, *p* < 0.001), beta-carotene (OR = 0.70, 95% CI = 0.56–0.88, *p* = 0.002), lutein and zeaxanthin (OR = 0.75, 95% CI = 0.59–0.95, *p* = 0.015) have showed the most statistically reduction in risk [112].

**Lycopene** is mostly found in tomato, watermelon, pink grapefruit, guava, and papaya [113]. Lycopene has been demonstrated to be a synergistic agent of cisplatin, able to increase HeLa cells’ sensitivity to cisplatin in vitro. Lycopene determined a significant enhancement of Bax expression and decreased Bcl-2 expression and has markedly activated NRF2 expression and suppressed the NF-κB signaling pathway [114,115].

**Saikasaponins** from *Bupleurum falcatum* sensitized HeLa and SiHa cells to cisplatin through intracellular ROS accumulation [116,117].

**Saponins** are mainly found in chickpeas and soybeans but are accumulated in several herbs, as well. In HeLa cells, saponins induced apoptosis by increased levels of cleaved caspase-3 and caspase-9, Bax, and Bcl-2 [118].

**Tanshinone IIA** from *Salvia* spp. has shown to be able to downregulate E6 and E7, induced apoptosis, cell cycle arrest, and inhibition of growth in HeLa, SiHa, CaSki, and C-33A cells [119]. In HPV positive cells, it has been shown to downregulate expression of HPV E6 and E7 genes and modulate associated proteins E6AP and E2F1, to cause S phase cell cycle arrest, induce accumulation of p53 and alter expression of p53-dependent targets, and modulate pRb and related proteins. In particular, authors have demonstrated that the tashinone IIA can cause p53-mediated apoptosis by moderating Bcl2, Bax, and caspase-3.

**Ursolic acid** can be found in apples, cranberries, peppermint, prunes, oregano, and thyme. Ursolic acid nanoparticles of poly(DL-lactide-co-glycolide) have been able to suppress cervical cancer cell proliferation, invasion, and migration through caspases and p53 in vitro in CaSki, HeLa, C4-1, and SiHa cells. In vivo, this compound has been able to reduce the size of CaSki, HeLa, and SiHa xenografts [120,121].

#### 4.2.2. Thiols

**Allicin** is mainly found in garlic and onion. In vitro studies conducted in SiHa cells have shown that allicin can induce apoptosis mainly by inhibiting the expression of NRF2 (nuclear factor erythroid 2-related factor 2) and increasing the level of glutathione and superoxide dismutase [122].

**Indole-3-carbinol (I-3-C)** can be found mainly in broccoli and brussels sprouts. Initial in vitro studies to test anti-cancer properties of I-3-C have been performed on C-33A cells and 17-beta-estradiol (E2)-treated mice have shown selective apoptotic potential on transformed cells and abolishment of the anti-apoptotic effects of estradiol, respectively [123].

These results have been upgraded by several clinical trials performed on I-3-C and its derivative 3,30-diindolylmethane (DIM, a stable form of I-3-C) (Appendix A, indole-3-carbinol-based formulations no.1–no.6).

A phase I trial, conducted in 1998 on 18 patients with recurrent respiratory papillomatosis, revealed that the use of I-3-C administered orally reduced the papilloma growth rate in 33% of patients (6 out of 18), resulting in cessation of papilloma growth and no surgery in 33% of them (6 out of 18), while no clinical response was found in the resulting 33% (6 out of 18) [124]. This phase I trial also established that I-3-C is safe and well-tolerated by children (Appendix A, indole-3-carbinol-based formulation no.1).

In addition, a placebo-controlled trial involving 27 patients diagnosed with CIN II–CIN III, resulted in a statistically significant complete regression in four out of eight patients orally supplied with I-3-C 200 mg/day and in four out of nine patients on I-3-C 400 mg/day. A change of the 2/16a-hydroxyestrone ratio was found (Appendix A, indole-3-carbinol-based formulation no.2) [125].

In 2004, an open-label clinical trial involving 33 patients with recurrent papillomatosis resulted in remission of papillomatous growth and no surgery in 11 (33%) patients treated with I-3-C and reduction in papillomatous growth and less frequent surgery in 10 (30%) patients, while there was no clinical response in 12 (36%) patients (Appendix A, indole-3-carbinol-based formulation no.3) [126].

Furthermore, a randomized double-blind placebo-controlled phase III trial (64 patients with CIN II or CIN III) demonstrated that CIN decreased by 1–2 grades or to normal histology in 47% (21) of the subjects on DIM treatment and that stratification by the level of dysplasia, age, race, HPV status, tobacco use, contraceptive use did not alter the results. This trial found no systemic toxicities as well (Appendix A, indole-3-carbinol-based formulation no.4) [127]. Conversely, a double-blind, randomized, controlled, primary prevention trial with 551 patients with low-grade cytological abnormalities demonstrated that DIM oral supplementation had no statistically relevant effect on cytology or infection compared to a placebo, with the development of CIN II or worse in 9% on DIM and 12% on placebo (risk ratio (RR) 0.7). Development of CIN III or worse (RR 0.9) occurred in 4.6% on DIM and 5.1% on placebo and no sign of disease in 27.3% on DIM and 34.3% on placebo at 6 months (RR 0.8) were found (Appendix A, indole-3-carbinol-based formulation no.5) [128].

Finally, a randomized, double-blind, multi-center clinical trial with a placebo was carried out to evaluate the effects and safety of 3,30-diindolylmethane (DIM, a stable form of I-3-C) administered as a vaginal suppository applied for 180 days for the treatment of CIN [129]. In total, 78 patients 18–39 years of age (56 out of 78 diagnosed with CIN I or CIN II) were enrolled. 200 mg/day and 100 mg/day determined a 100% (CI 95% = 82.35–100.00%) and 90.5% (CI 95% = 69.62–98.83%) regression of CIN lesions, respectively with significantly lower regression in the placebo group (61.1%, CI 95% = 35.75–82.70%) (Appendix A, indole-3-carbinol-based formulation no.6).

**Sulforaphane**, a isothiocyanate of cruciferous vegetables [130] in HeLa, Cx, and CxWJ cervical cancer cell models has shown dose-dependent cytotoxicity associated with arrest in G2/M phase through Cyclin B1 downregulation [131].

### 4.3. Miscellaneous Specialized Metabolites

**Berberine** (benzylisoquinoline alkaloid) mainly accumulated by plants of the *Berberis* genus and possesses general anti-inflammatory and anti-cancer properties through inhibition of AP-1 transcription factor with no known toxicity. In SiHa and HeLa cells, berberine specifically downregulates the expression of oncogenic c-Fos [132]. Moreover, berberine has been able to reduce expression of E6 and E7 with a concomitant increase in p53 and pRb expression, loss of telomerase protein, hTERT, resulting in growth inhibition of cervical cancer cells [132]. Furthermore, in HeLa cells, berberine has been shown to alter epigenetic modifications and to disrupt the microtubule network by targeting p53 [133].

**Betaines** are common in beetroot and have demonstrated proliferation inhibition of HeLa cells through G1/S or S/G2 cell cycle arrest and dose-dependent apoptosis induction through Bax, p53, caspase-3, and Bcl-2 modulation [134].

**Decursin and decursinol** (pyranocoumarin compounds) from the roots of *Angelica gigas*, were able to make HeLa cells more sensitive to tumor necrosis factor-related apoptosis-inducing ligand (TRAIL) by activating caspase-8 and caspase-9 [135].

**Piperine** (1-pipeoylpeperdine) can be found in black and long peppers (*Piper nigrum*, *Piper longum*). Endowed with anti-cancer activities, it has been tested against HPV models as well. In HeLa cells, it was demonstrated to induce dose-dependent apoptosis through caspase-3, DNA fragmentation, ROS generation, and nuclear condensation [136].

**Withaferin A** (steroid lactones) from *Withania somnifera* induced apoptosis of CaSki cells through E6 and E7 downregulation and consequent inhibition of tumor growth [137]. In particular, withaferin A alters expression levels of p53-mediated apoptotic markers Bcl2, Bax, caspase-3, and cleaved PARP.

### 4.4. Polysaccharides

**Crude polysaccharides** from *Glycyrrhiza uralensis* have been recently investigated as adjuvants of HPV DNA vaccine tested in a preclinical model to detect the efficacy of combinatorial approaches in improving antigen-specific cellular and humoral immune responses. C57BL/6 mice were co-immunized with HPV-DNA vaccine and polysaccharides. Crude polysaccharides enhanced the DNA vaccine-induced antigen-specific CD4+ and CD8+ T cell responses and the levels of IgG, IgG1, and IgG2a. Researchers also demonstrated that administration of *Glycyrrhiza* crude polysaccharides had no adverse effects in mice but significantly increased B cells and macrophages in the spleen. Intragastric administration of polysaccharides also significantly increased the levels of CD4+ and CD8+ T splenocytes [138].

### 4.5. Proteins and Peptides

**Lectins** from different species among which *Clematis montana* and *Astragalus mongholicus* have been tested against HeLa cells in vitro showing antiproliferative properties through induction of apoptosis via caspase-dependent pathways and S-phase arrest, upregulation of p21 and p27, and reduction in active complex cyclin E/CDK2 kinase [139].

**Peptides** from *Triticum aestivum*, *Abrus precatorius*, and *Trichosanthes kirilowii* have been shown to have antiproliferative activities in HeLa cells in vitro by different mechanisms such as induction of DNA damage and G2 arrest and induction of apoptosis with the generation of ROS [140,141,142].

## 5. Combinations of Purified Phytocompounds

Specialized metabolites, as purified and characterized entities used in combination, are reviewed in this section.

**Alpha–, beta–, and gamma–carotene and zeaxanthin carotenoids**, previously studied as individual compounds [108,110], have been evaluated in combination case studies, as well, demonstrating that the risk of developing cervical cancer is inversely correlated with blood levels of this combination of compounds in patients [143].

**Curcumin and ellagic acid** combined at different concentrations have shown improved inhibitory effects on HeLa cells compared to the single compounds administered individually by MTT assay. The combination of curcumin and ellagic acid restored p53, increased ROS levels, and induced DNA damage as well. It has been further demonstrated that curcumin and ellagic acid determined a downregulation of the HPV E6 oncoprotein as well [144].

**Purified green tea compounds** have been tested in different clinical settings.

Different ECGC-containing (200 mg) formulations, such as polyphenon E (poly E, decaffeinated and enriched green tea catechin extract, containing 85–95% total catechins, with 56–72% as EGCG) and/or ECGC capsules, have been evaluated in 90 patients with HPV cervical lesions from chronic cervicitis to severe dysplasia. Clinical effects have been proven both through intravaginal and oral delivery of the two formulations (on a daily basis, for 8–12 weeks). Indeed, 74% of patients treated topically with poly E ointment, 75% of patients treated both topically and orally with poly E ointment + oral EGCG, 50% of patients treated orally with poly E, and 60% of patients treated orally with EGCG, showed a significant ameliorating response (Appendix A, catechin-based formulation no.1) [145].

A phase II randomized, double-blind, placebo-controlled trial (98 patients with persistent high-risk HPV infection and CIN I, daily treated for 4 months with oral poly E capsules), demonstrated no statistically relevant promotion of neither clearance of persistent high-risk HPV or normal histopathology but demonstrated that poly E is acceptable, safe, and well-tolerated (Appendix A, catechin-based formulation no.2) [146].

**Rutin** (a flavonoid common in capers, olives, buckwheat, asparagus) purified from *Citrus sinensis* peel was combined with the **fucoidan polysaccharide** from *Fucus vesiculosus* algae in a rutin-fucoidan (Ru-Fu) to form a complex with low bioavailability of flavonoids. The Ru-Fu complex has been shown to effectively induce anti-proliferation, S-phase cell cycle i and apoptosis in HeLa cells via nuclear fragmentation, ROS generation, and loss of mitochondrial membrane potential [147].

## 6. Plant Extracts

The use of fractions or extracts from plant parts may represent an easier and cheaper approach compared to highly purified phytocompounds. For this reason, extracts of different origins and types have been tested for anti-HPV properties.

Extracts from the stem of *Cudrania tricuspidata* have been shown to induce dose-dependent cytotoxicity in SiHa tumor cells, but not in HaCaT keratinocytes at 0.125–0.5 mg/mL concentrations. The mechanism underneath this selective effect has been related to downregulation of E6 and E7 and apoptosis triggering related to upregulation of Fas, death receptor 5, TRAIL, activation of caspases-3 and -8, and cleavage of PARP (poly-ADP-ribose polymerase) [148].

*Ficus carica* fruit latex has been demonstrated to inhibit the rapid growth and invasiveness of CaSki and HeLa cells through downregulation of p16 and E6 and E7 gene expression [149].

Aqueous extract from *Crocus sativus* and its two main purified compounds (i.e., picrocrocin, a monoterpene aldehyde; and the carotenoid crocin) reduced TC-1 cell proliferation in vitro in a concentration- and time-dependent fashion. When orally administered in association with a DNA vaccine encoding E7-NT (gp96), both extracts and purified compounds acted synergistically in the inhibition of experimental tumor growth in mice, acting either as preventative (aqueous extract, picrocrocin) or therapeutic (crocin) agents [150].

## 7. Plant Extracts Mixtures

Extracts of different origins have also been tested in combination against HPV to possibly integrate the benefits of multiple components.

Inhibition of AP-1 and STAT3—which are known to promote cervical carcinogenesis, and specifically downregulate the expression of viral oncogenes E6 and E7—has been found to represent the major mechanism of action of a mixture of extracts from *Pinellia pedatisecta* (rhizome extract), *Bryophyllum pinnata* (fractionated leaf extract, enriched in bryophyllin A), and *Phyllanthus emblica* (fruit, crude extract), *Brucea javanica* (oil emulsion) in different human cervical cancer in vitro models [151,152,153].

**Basant**, a mixture of extracts from *Emblica officinalis* and *Aloe vera*—added with curcumin, saponins, and *Mentha citrata* oil and Praneem—blocked transduction of HPV-16 pseudovirus in HeLa cells, when diluted at non-cytotoxic concentrations (IC_50_ 1:20,000). Basant was found to be safe according to pre-clinical toxicology carried out on rabbit vagina after application for 7 consecutive days or twice daily for 3 weeks [154].

Therefore, a placebo-controlled phase II randomized study in which either intra-vaginal application of curcumin-containing capsules or basant ointment was performed in 280 HPV-infected women in absence of high grade CIN. This study has demonstrated a statistically relevant resolution of cervical HPV infection in the case of basant ointment application for 30 days and no severe adverse events have been reported (Appendix A, curcumin-based formulation no.2) [155].

All the 11 HPV-infected women with low-grade cervical lesions enrolled in a placebo-controlled phase I trial with basant, cleared HPV-16 infection upon basant application (Appendix A, curcumin-based formulation no.3) [156].

**Praneem** polyherbal, a mixture of extracts from *Azadirachta indica* and *Emblica officinalis*—combined with saponins, *Mentha citrata* oil, and *Aloe vera* ge—have been examined in 20 women diagnosed with low-grade squamous intraepithelial lesions in a placebo-controlled clinical study [157]. Topical administration (500 mg, intra-vaginally, 30 day-treatment) cleared HPV-16 in 60% of patients. A subsequent treatment was administered to women who had not cleared the infection, determining the end of infection in 80% of patients overall (Appendix A, other polyherbal-based formulation no. 1). Microbicidal action against infectious agents that are known to cooperate in the promotion of HPV carcinogenesis has been demonstrated for Praneem and may disclose further effects on cervical lesions [158].

## 8. Phytocompounds with Anti-HPV Potential Inferred by In Silico Approaches

With the evolution of sophisticated molecular modeling, in silico approaches have been utilized to target HPV oncoproteins to make them non-functional.

As a consequence of these studies, it was inferred that daphnoretin (a protein kinase C activator isolated from the plant *Wilkostroemia indica*), withaferin A, and artemisinin (a sesquiterpene lactone derived from *Artemisia annua*) have docking capacity on E6 from HPV-16 and HPV-18 and may exert an inhibitory effect nearly 20 times stronger than curcumin, ECGC, or jaceosidin, which are known to functionally interfere with E6 [159,160].

Furthermore, 20 plant-derived compounds from different sources have been collected by Nabati and colleagues and tested against E6 [161]. In this study, the molecular interaction of HPV-16 E6 protein with plant-derived inhibitors in the proximity of E6AP, p53, and Myc binding sites were investigated using docking analysis. Ginkgetin, hypericin, and apigetrin have shown significant ability to block three binding sites on HPV-16 E6 oncoprotein with minimum binding energy [161].

With the aim to get a stronger immune response against HPV infection, an in silico model to predict the adjuvant activity of plant-derived compounds in the context of preventive immunization was built as well. This study predicted that neohesperidin, a flavanone glycoside mainly found in citrus fruits, may be able to elicit an immune response and subsequently validated this hypothesis in vivo in a mouse model. This integrated methodology (virtual screening combined with in vivo testing) allowed us to set up a protocol to confer humoral immune response against HPV infection [162].

Despite the power, speed, and potential of this approach, further studies and translations into experimental settings are needed to validate findings and to develop rationally designed drugs.

## 9. The (Plant) Cell Factory

As highlighted, plant cells represent a natural factory of useful metabolites. For pharmaceutical production ends, when naturally occurring sources of a compound of interest are rare, endangered, available only on a seasonal basis, or when these molecules need qualitative or quantitative improvement, ‘artificial’ production in a bioreactor can be accomplished. Microbes, such as bacteria and yeasts—but especially plant-based production platforms—can be used to this end. The use of plants as biofactories for the production of either small molecules (i.e., specialized metabolites, sugars, and others) or complex polymers like proteins (such as lectins) that may be either natural or exogenous is known as ‘plant molecular farming’ (PMF; Figure 3) [163,164,165]. Different bioreactor formats have been derived from plants to this end: whole plants, cell suspension, and organ cultures. Plant cell totipotency and plasticity, low-cost upstream production, intrinsic safety based on the inability of human pathogens to replicate in plant cells, and potential for industrial large-scale production with manufacturing capacity rapidly scalable to meet market demands, are benefits generally applicable to plant-based production platforms making PMF attractive for future applications [165,166,167].

In the case of small molecules, once the biosynthetic pathways responsible for the biosynthesis of a desired product and its regulation by the involved enzymes and transcription factors are clarified, the expression of one or more sets of the necessary enzymes and/or transcription factors can be accomplished. Stable expression in transgenic whole plants, plant cells, and plant organs has been the main expression strategy applied to PMF of specialized metabolites and it has been accomplished in natural hosts or in exogenous species if technologically feasible [168] (Figure 3).

The reconstitution of the flavonoids pathway was demonstrated in both *E. coli* and *S. cerevisiae* [169,170]. Nevertheless, modifications—such as glycosilation, methylation, and acylation—that are needed to obtain more ‘authentic’, stable, and biologically active flavonoids have to be assured to occur in the requested pattern in these systems. Engineered microbial systems have also been demonstrated to be useful to produce derivatives of flavonoids that do not occur in nature, and these compounds are being investigated for their pharmaceutical potential [171]. Carotenoids, akaloids, betalains, and glucosinolates are among the further specialized metabolites that have been mostly approached by metabolic engineering [172].

Especially regarding the production of proteins with anti-cancer activity, many plant species and the different culture formats thereof can produce them naturally or have been used as expression platforms for recombinant, exogenous production [165]. Plant cells can synthesize proteins with complex structure, preserving their authentic post-translational modifications (e.g., glycosylation, disulfide bond formation) and eventually improving function (‘biobetter’ pharmaceuticals). An application of these advantages occurs in the plant-based production of eterodimeric lectins such as viscumin. Lectins are anti-tumoral complex glycoproteins that also have proven effectiveness against HPV in vitro. These are ‘familiar’ compounds for plants and heterologous lectins are commonly less toxic for plant cells than for other cell types. In plants, viscumin can be produced from a single transcript and the monomeric precursor can be processed as in the native host, which further lowers process complexity. Otherwise, lectins should be produced by separate bacterial fermentations and should require out-of-bioreactor refolding and glycosilation (necessary for stability and efficacy), leading to poor recoveries. On the other hand, production by yeasts would decorate lectins with high-mannose rather than complex-type glycans [173,174,175]. Mammalian cells are unlikely to express lectins at high levels because they would be toxic [176].

## 10. Marine Substances

A few marine-derived bioactive compounds have been shown to have anti-HPV and related cancer activities, demonstrating the anti-HPV potential of marine resources ‘competing’ with the well-known plant-derived compounds. Polysaccharides, among other bioactive compounds, present in different marine organisms—from microbes to algae to invertebrates—have been examined for potential application against HPV.

### 10.1. Algae

**Carrageenan**, a sulfated polysaccharide of D-galactose extracted from red algae, is widely used as a thickener in cosmetics and food products [177]. Carrageenan has been demonstrated to act by impairing the binding of HPV to cells and, subsequently, by blocking infection through heparin sulfate-independent mechanism [177]. This ability was maintained by milk-based products containing carrageenan, even though this was at a lower efficiency than pure carrageenan [177]. Carrageenan has been shown to block HPV genital transmission in a mouse cervicovaginal model as well. The sulfated galactose structure and content have been defined as important factors for the anti-HPV effect of carrageenans [178].

**Fucan** (i.e., L-fucose-containing sulfated polysaccharides) contained in a polysaccharide-rich extract from the brown seaweed *Sargassum filipendula* showed significant reduction in HeLa cell proliferation by release of apoptosis-inducing factor (AIF) into cytosol from mitochondria [179]. In another work, an isolated antioxidant heterofucan (SF-1.5v) inhibited HeLa cell proliferation by a molecular mechanism that remains unclear [180].

**A fucose-enriched fraction**, isolated from the brown seaweed *Sargassum stenophyllum* has been shown to significantly alter cellular morphology of HeLa cells and to reduce cell growth in a dose-dependent manner (40–80 µg/mL) [181].

**Heparin and heparinoids** are sulfate polysaccharides that have been demonstrated to interfere with the initial attachment of HPV viral particles to the host cell [182]. Through the same mechanism, the marine heparinoid polysaccharides, alginic acid and fucoidan, from brown algae have been demonstrated to effectively block HPV pseudovirions infection [177]. In addition to blocking infection, alginic acid from the brown seaweed *Laminaria brasiliensis* demonstrated an inhibitory effect on HeLa cells by promoting atypical mitoses and nuclear fragmentation [181].

### 10.2. Marine Invertebrates

**Chitosan**, which can derived from deacetylation of chitin from the shells of crustaceans, has been demonstrated to have inhibitory effects on HPV infection-induced chronic cervicitis with a superior negative conversion rate compared to interferon α-2b gel [182]. Diethylaminoethyl chitosan has been found to trigger apoptosis in HeLa cells through upregulation of caspases, p53, and Bax, and downregulation of Bcl-2 [183].

**Hemocyanins** are a large family of respiratory glycoproteins found in some mollusks. They possess several beneficial effects linked to potent Th1 adjuvant activity. Keyhole limpet hemocyanin (KLH) and two new hemocyanins from *Concholepas concholepas* (CCH) and *Fissurella latimarginata* (FLH) have been utilized in a pre-clinical mouse model of orthotopic oral HPV-associated cancer. In order to potentiate the antitumor activity, the two hemocyanins were applied in combination with QS-21, a synthetic saponin. The KLH and FLH in combination with the QS-21 adjuvant showed reduced tumor development and greater overall survival. This study suggests potential for the effective use of hemocyanins plus adjuvants for the development of immunotherapies against head and neck cancers [184].

**Manzamine A**, an alkaloid from an Indo-Pacific sponge, has been demonstrated to have antiproliferative activity against C33A, HeLa, SiHa, and CaSki cells [185]. In SiHa and CaSki cells, the antiproliferative effect of manzamine has been demonstrated to occur at non-cytotoxic concentrations (up to 4 μM) through the block of cell cycle at G1/S phase and by regulation of cell cycle-related genes and restoration of p21 and p53. Along these anti-tumor properties, interestingly, manzamine A administration led to a significant decrease in SIX1 and CK2α proteins intracellular level along with the regulation of cell-cycle related checkpoint proteins in the mentioned cell lines, suggesting possible activity in vivo against cervical cancer, since it expresses high levels of SIX1. In a direct comparison to apigenin, a known inhibitor of cyclin-dependent kinases, manzamin demonstrated a 10-fold inhibiting activity on CK2α and SIX1 proteins than apigenin.

**Mer2**, a low molecular weight polypeptide from the clam *Meretrix meretrix* has been demonstrated to significantly block the growth of HeLa cells through a dose- and time-dependent mechanism and has induced morphology changes and apoptosis [186].

**SJAMP**, a polysaccharide isolated from the marine echinoderm *Apostichopus japonicus*, have been shown to significantly inhibit the proliferation of HeLa cells [187]. In this model, SJAMP downregulated the proliferating cell nuclear antigen (PCNA) and the cell cycle inhibitor protein Mdm2, and induced the arrest of cell cycle in G1 phase [187].

### 10.3. Marine Microbes

Marine microbes can be sources of different bioactive compounds with anti-HPV cancer activity as well.

**Antimycin alkaloids** and related analogs isolated from the lab-grown marine *Streptomyces* sp. THS-55 has exhibited cytotoxic activity in vitro against HeLa cells. One of them in particular, denoted as NADA, showed the highest potency by inhibition of proliferation, cell cycle arrest, and apoptosis induction. NADA has been shown to degrade HPV E6 and E7 viral oncoproteins and to inhibit their function through ROS-mediated ubiquitin-dependent proteasome system activation [188].

**Gliotoxin**, which can be produced by marine *Aspergillus* species, has reduced the proliferation of HeLa cells by apoptosis associated with a drop of mitochondrial membrane potential and activation of Bax, caspase-3, caspase-8, and caspase-9, together with suppression of Bcl-2 [189].

**Neoechinulin A** (prenylated indole alkaloid) produced by cultured marine-derived *Microsporum* sp. [190] has shown cytotoxic effect on HeLa cells through downregulation of Bcl-2 and upregulation of Bax and activating the caspase-3 pathway.

Despite the potential to block HPV infection, as well as to have in vitro effects, these marine-derived compounds need to be explored and characterized further. Current research suggests that marine natural products of proven efficacy show low cytotoxicity, low production costs, and acceptability as novel drug tools [191,192].

## 11. Bacterial Substances

There is evidence that, in one case, bacterial compounds have effects on HPV. HPV-16, -18, and -6 pseudovirions have been inhibited by *E. coli* K5 capsular polysaccharides due to their similarity with the backbone of heparin/heparan biosynthetic precursor [193].

## 12. Discussion

HPV-related lesions still represent a major concern for public health. The number of recorded cancer cases is likely to increase due to demographic changes and to improved and more widely available diagnostics. Combined with increasing access to healthcare systems in developing countries, this will enhance the demand for anti-HPV agents. More importantly for developing countries, approaches satisfying criteria such as to be possibly self-applied, non-destructive, with minimal side effects—while being effective and affordable—should be fulfilled to improve pharmacological treatment of HPV. Reducing side effects of currently available radio- and chemo-therapy is central as well. Therefore, exploring additional approaches to HPV prevention and therapy is useful to possibly corroborate current intervention methods.

Plants are the main source of chemicals that demonstrated effects against HPV. Despite the ubiquitous polyphenols receiving the most attention in in vitro/pre-clinical studies, several different mechanisms are exerted by natural compounds against HPV in various models: apoptosis induction, growth arrest, effects on DNA, alteration of signal transduction, alteration of redox balance, as well as capacity to specifically modulate the expression and activity of HPV oncogenes. These mechanisms play roles in different steps of the cascade of HPV-induced cell transformation and indirectly show the possible contribution of these compounds in the fight against HPV.

One encouraging feature of plant-derived dietary compounds is their behavior as safe and effective chemo- and radio-sensitizer agents. A possible drawback is the eventual poor bioavailability of some of these compounds. To improve this aspect, encapsulation or complexation of plant compounds with other (bio)materials to form nanoparticles or complexes, respectively, has been considered. Indeed, it has been demonstrated that complexes may improve stability and, as a consequence, targeted release. On the other hand, nanosized phytochemicals have demonstrated high stability, improved absorption rates, and reduced compound degradation as well. Nevertheless, nanotechnology remains an issue for clinical success, mainly due to safety.

Considering the different and peculiar targets of action of each natural compound that has been tested against HPV, the synergistic use of molecules may be considered as a tool to amplify their effects on HPV. The few, recent studies available on the improved effects of association of purified phytochemicals compared to separate entities seem to point in this direction. In this sense, purified phytochemicals, alone or in combination—due to wider availability, precise identity, quantity and batch-to-batch reproducibility—may probably represent the safest and the most feasible entities to be translated to clinics. Other formulations—such as extracts, suffering from batch-to-batch changes—may be unfeasible for real clinical applications. Phytocompounds derived from food and traditional medicine may have an advantage over others, due to use-proven non-toxicity within certain ranges. Compounds that directly affect HPV E6 and/or E7 are probably the most promising candidates for HPV therapy, due to their impact on the major and most specific players of HPV oncogenesis.

Despite plants remaining the main source of compounds with effects on HPV, notably, a few studies on marine compounds of different origins (algal, microbial, animal) are available, and they offer examples that further enlarge the number of compounds with pharmacological values against HPV.

Undoubtedly, additional basic research to further evaluate the properties and effects of natural (and, especially, plant-derived) compounds is needed—in vitro and in vivo—to better clarify mechanisms of action and to evaluate the entity of the interference with cell pathways. Studies on the effects on the immune system are lacking. These studies may be of significance to investigate if natural compounds may have a role in modifying the immunosuppressive tumor microenvironment as well. It would be also desirable to study and delve into the role of natural bioactive molecules in persistent HPV infection, although diagnostic methods for detecting HPV viral latency are still lacking. Natural products, and especially phytochemicals, may help in this area. Immune-enhancing natural compounds that can potentiate innate or antigen-specific cell-mediated immune responses should be taken into consideration. Results from the few studies available suggest that combinatorial treatment strategies, such as combining immunotherapy with the administration of a natural compound with tumor-killing properties, may be a more effective anticancer strategy than single-modality of treatments.

Several clinical trials (mostly phase I–II) have been published demonstrating the possible clinical use of phytocompounds (listed also in Appendix A), mostly with no concerns regarding safety issues, and some of them with encouraging results. Apart from the outcome, ablative procedures on advanced HPV lesions require specialized infrastructure or expensive approaches. Natural compounds would represent a simpler, safe, and cost-effective approach with extensive pre-clinical validation. They would be easily applied both orally and topically if clinical elimination of HPV infection or reversion of lesions could be effectively and finally proven. These features would make them well suited to application in low-income developing countries. To this end, more structured, uniform, large-scale, and multi-centered clinical trials than those that have been already reported are strongly needed. These are crucial to provide the necessary information for a clear assessment of their human use.

Among the most clinically studied (at least in the western countries) substances of natural origin, DIM appears promising. As already introduced, DIM is a naturally occurring indole found in cruciferous vegetables. It is a well-known dietary supplement. DIM is derived from the transformation of indole-3-carbinol (I-3-C) in the acidic environment of the stomach. The chemopreventive, anti-mutagenic, and anti-carcinogenic properties of DIM and I-3-C also have potential against HPV. As a derivative of I-3-C, DIM reduces the levels of 16-hydroxy estrogen metabolites and increases the formation of 2-hydroxy metabolites, thereby increasing antioxidant activities. Clinical reports of success have been published in the treatment of HPV-related lesions of the cervix and the larynx with this family of compounds (Appendix A). Despite mostly favorable data, some of these studies failed to demonstrate a statistically significant advantage in the use of these compounds. Sometimes, this may be explained with the wrong choice of the placebo (i.e., Del Priore et al. [131] used rice bran that may have potential activity against HPV lesions). Nevertheless, it is difficult to arrive at a final conclusion due to lack of uniformity in dosage and administration route (DIM intravaginal route seem to be the most promising in the case of cervix lesions). In order of importance (i.e., as per the number of published clinical studies), curcumin-derived compounds follow those based on DIM and I-3-C, especially in Eastern countries. Nevertheless, these studies suffer from either a very low number of enrolled patients or, when enrolling wider patient numbers, lack full statistical relevance. In addition, despite not directly concerning HPV, a study carried out on breast cancer patients revealed that the use of curcumin could reduce tamoxifen and endoxifen concentrations below the levels of therapeutic efficacy, suggesting that tamoxifen treatment efficacy should be adequately monitored in these patients [194]. This study highlighted the importance of monitoring the activity of bioactive compounds on concomitant treatments with other drugs.

Available HPV clinical data, regardless of the compound and formulation, seem to point out to a general safety of phyto-compounds. However, clinical research has still to fill the current gaps of knowledge about optimal doses and pharmaco-dynamics. Clinical research should also focus on a more detailed evaluation of the interactions of these compounds with other drugs and possible adverse effects for comparative studies with canonical treatments and within combinational schedules. Further investigation on the synergistic effects among compounds with approved HPV therapy schedules should be accomplished. The latter approach may be the most interesting application of phytochemicals if improvement of action and mitigation of adverse effects of canonical radio- and chemo-therapy-based approaches could be proven.

In the absence of highly finalized clinical research, the wealth of currently available studies runs the risk of providing only information limited to the huge variety of compounds, formulations, and targets that have been tested so far for potential against HPV. All the work done will remain necessarily associated with strong limitations until the most promising compounds are chosen for conclusive clinical investigation and possibly authorized for safe human use.

Once these goals will be achieved, thanks to current knowledge in the regulation and control of metabolic networks in plants and bacterial biofactories, both synthetic biology and scalable industrial production of natural compounds could be guaranteed and qualitatively/quantitatively improved for wide clinical application. The use of plant biotechnology and its advances will especially be of help in delivering such agents, since established manufacturing platforms based on mammalian cells and microbes are endowed with either high production costs, anti-cancer-agent inherent toxicity, or synthesis quality limitations.

## Figures and Tables

**Figure 1 jcm-11-01465-f001:**
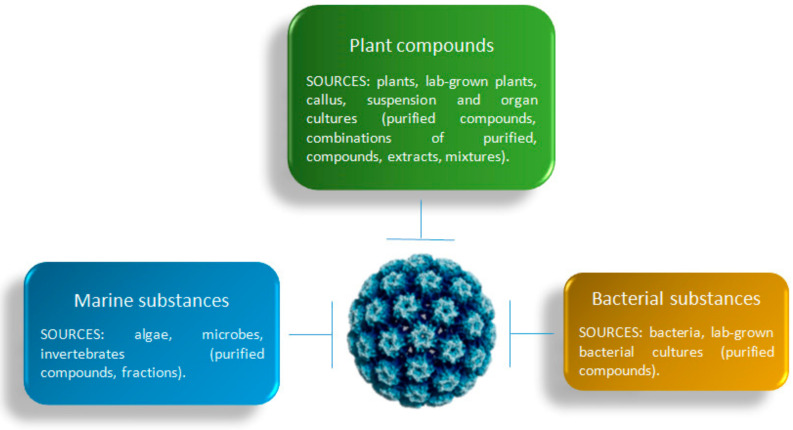
Sources (plants, marine environment, bacteria) and nature (purified compounds and their combinations, fractions, extracts, mixtures) of formulations that have been investigated against HPV.

**Figure 2 jcm-11-01465-f002:**
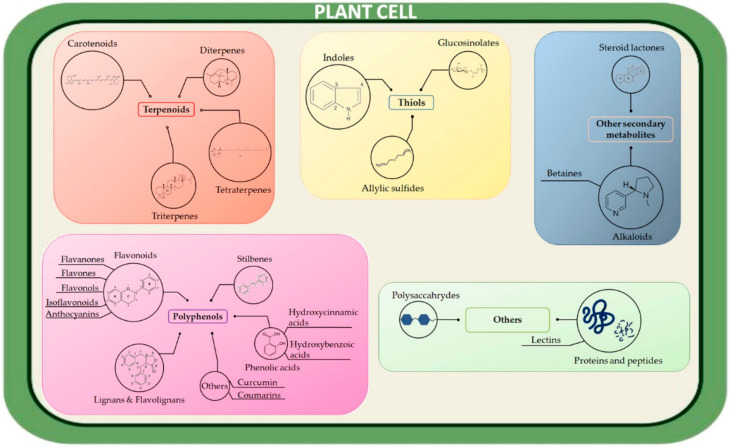
Phytochemicals investigated against HPV.

**Figure 3 jcm-11-01465-f003:**
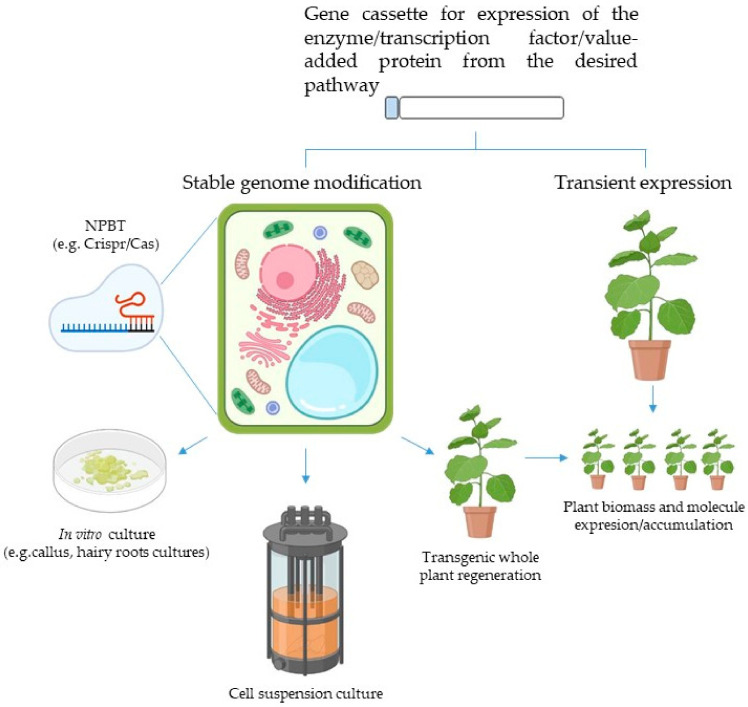
The plant molecular farming approach. Through stable or transient expression of recombinant proteins, either exogenous proteins and metabolites accumulation or improvement of biosynthesis pathways can be achieved (NPBT: new plant breeding techniques; Crispr-Cas: clustered regularly interspaced short palindromic repeats–Crispr-associated protein).

**Table 1 jcm-11-01465-t001:** Literature search details.

Database/Platform:	MEDLINE (PubMed)
Date coverage:	From 1995 to present
Library:	National Library of Medicine, National Center for Biotechnology Information
Date of last search:	20 October 2021
Limits:	In: “Article Title, Abstract, Keywords” Published: “All years” to “Present” Document type: “All” Subject Areas: All checked (default)
Search query:	HPV disease + natural compounds + bioactives + phytochemicals + combination treatments + therapy + infection + name of compounds/class of compounds
Number of hits:	22
Database/Platform:	Science Direct (Elsevier)
Date coverage:	From 1998 to present
Library:	Free access
Date of last search:	20 October 2021
Limits:	In: “Article Title, Abstract, Keywords” Published: “All years” to “Present” Document type: “All” Subject Areas: All checked (default)
Search query:	HPV disease + natural compounds + bioactives + phytochemicals + combination treatments + therapy + infection + name of compounds/class of compounds
Number of hits:	2870
Database/Platform:	Scopus (Elsevier)
Date coverage:	From 1995 to present
Library:	Free access
Date of last search:	20 October 2021
Limits:	In: “Article Title, Abstract, Keywords” Published: “All years” to “Present” Document type: “All” Subject Areas: All checked (default)
Search query:	HPV disease + natural compounds + bioactives + phytochemicals + combination treatments + therapy + infection + name of compounds/class of compounds
Number of hits:	20
Database/Platform:	Google Scholar
Date coverage:	From 1995 to present
Library:	Free access
Date of last search:	20 October 2021
Limits:	In: “Article Title, Abstract, Keywords” Published: “All years” to “Present” Document type: “All” Subject Areas: All checked (default)
Search query:	HPV disease + natural compounds + bioactives + phytochemicals + combination treatments + therapy + infection + name of compounds/class of compounds
Number of hits:	1330

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
