# Peer review of "Natural Bioactives: Back to the Future in the Fight against Human Papillomavirus? A Narrative Review"

_jcm, 2022, doi:10.3390/jcm11051465_

Round 1

Reviewer 1 Report

Thanks to the authors for this new version of their article which is much better. The review is well structured and seems to me complete and up to date.

However, even if this article is a narrative review, it does not prevent the authors from adding a search method part, which describes precisely the databases consulted, and on what criteria the articles were selected and analyzed in order to be included in this review.

Author Response

We thank the Reviewer for the comments. We have modified the section to provide the required information.

Reviewer 2 Report

The authors have responded to one of my comments that they were unable to identify any previous studies that may have suggested that curcumin consumption may negatively impact the efficacy of chemotherapeutic agents. For the benefit of the authors, I am sharing the following link  https://doi.org/10.3390/cancers11030403.

This is the a link to a published study that compared the efficacy of tamoxifen (a chemotherapeutic agent widely used for the treatment of breast cancer) and endoxifen (the active metabolite of tamoxifen) in breast cancer patients who were administered tamoxifen with or without curcumin and piperine. The authors observed that co-administration of curcumine with tamoxifen could significantly reduce the concentrations of tamoxifen, as well as endoxifen - curcumin co-administration could even reduce their concentrations below the levels of therapeutic efficacy.

Since the role of bio-actives in the treatment of cancers is an area under development, a review on this topic should present these various facets and provide a more comprehensive overview of conflicting results obtained in different studies to facilitate a more nuanced discussion.

The authors could consider discussing this in the manuscript.

Author Response

We thank the Reviewer for review and for the link to the work that he was intended to indicate. We added results and reference of this paper in Discussion, as indicated.

Reviewer 3 Report

Dear authors,

please find attached comments.

Author Response

The manuscript Natural Bioactives: Back to the Future in the Fight against Human Papillomavirus? A narrative review by Massa et al. covers interesting topic.

The manuscript needs substantial editing. Please be consistent – when the authors use an abbreviation, use it consistently; authors should decide between words with or without hyphens (line 576 down-regulation vs line 580 downregulation), lowercase or uppercase (line 416 Nrf2 vs line 435). The authors should explain abbreviations when they are first introduced. It is very distracting to review a poorly prepared manuscript.

We thank the Reviewer for this comment. We have introduced the suggested modifications throughout the text.

The authors should be aware of the focus of their own manuscript. It is important to distinguish between the antitumor properties of a substance or compound and the anti-HPV properties, which is the topic. For example, if a substance such as manzamine A acts against C-33A and HeLa (lines 786-787), it can be concluded that this substance has antiproliferative properties against tumor cells, it is not anti-HPV specific.

We have checked all the compounds that have been tested against both HPV-positive and HPV-negative cell models (i.e. EGCG, Curcumin, Tanshinone II A and Manzamin A) and we have accordingly improved the text to better reach the scope of the review and to distinguish HPV-specific results from general anti-tumor effects. We have erased reference to those compounds that have been tested only against HPV-negative cervical cancer cell models (Silymarin).

Introduction

Line 43: delete ‘…with different percentages'

Done

Lines 38-47: reorganize the paragraph

‘…As it is well known, HPV tumors mainly originate from persistent infection with high-risk types HPV-16 and HPV-18, that are linked to the persistent infection of oncogenic strains of HPV [6,7]. However, HPV infection is also linked to anogenital cancers (cervical, vulvar, vaginal, anal, and penial), genital warts or and head and neck cancers (HNCs) in both men and women with different percentages [4,5]. Among HPV-associated tumors, cervical cancer is still ranking fifth in terms of both incidence (6,5% of total female cases) and mortality (7,7% of total female deaths for cancer) from all cancers in the female population [5]. Approximately all cervical cancers are associated to HPV infection. Importantly, 50% of all HPV-associated cancers in women are cervical cancers, whereas in men nearly 80% are oropharyngeal cancer [4].

Done

There is no need to introduce abbreviations if they are used only once (HNC).

Ok, thanks.

Line 55: correct Rrimantadine

Done

Line 58: correct intra-bodies

Done

Line 79: '...reducing/preventing/inhibiting...' use ',' instead of '/'

Done

Line 86: countries instead Countries

Done

Authors should reconsider this sentence 'Dietary bioactives might be especially beneficial in those Ccountries where only limited or no access to highly specialized treatment is possible' as this could imply that dietary bioactives are a cheap, potentially less effective variant.

We agree and we deleted the sentence.

Line 95: anti-oxidant without '-'

Done

Line 103: bio-actives without '-'; instead ‘…natural origins...' use sources

Done

Line 104: delete '... throughout the different natural sources.'

Done

Lines 116-122: delete paragraph

We agree and we deleted the sentence.

Lines 127-136: this paragraph should be under Methodology section

Done

Methodology

Please provide more information: which keywords authors used, specify the date when each source was last searched or consulted.

We have modified the section to provide the required information.

Line 143: add / ‘…and or research papers...'

In the modification of the methodology section, this part has been deleted and rewrited according to Reviewer’s suggestion.

Line 144: '...in vitro/ in vivo/case/clinical randomized...' instead of '/' use ',' as separator

In the modification of the methodology section, this part has been deleted and rewrited according to Reviewer’s suggestion.

Authors should include Result section where they will describe the results of the search and selection process, from the number of records identified in the search to the number of studies included in the review, which period was included, articles’ language, how many articles was excluded

We have modified the text according to the Reviewer’s suggestion (i.e., a Results section has been added and the numbering of paragraphs has been accordingly modified).

Lines 152-160: unnecessary paragraph

Done

Line 173 section 3.1.1. Phenolic acids are not clearly and obviously connected with Figure 2. only for gallic acid authors emphasized that it is hydroxybenzoic acid

We have now specified the chemical subclass in the text, so that now both hydroxycinnamic and hydroxybenzoic acids cited in the text are clearly related to Figure 2.

Line 180: introduce abbreviation ROS; put the abbreviation in parenthesis and the full name in front of it - histone deacetylase (HDAC).

Done

Line 185: put the abbreviation in parenthesis and the full name in front of it

Done

Line 188: correct bio-availability

Done

Line 190: correct Hela

Done

Line 198: explain GSH

Done

Line 200: put the abbreviation in parenthesis and the full name in front of it

Done

Line 203: italicize vivo

Done

Lines 218 and 229: please use the same designations for the same thing (p21WAF1 vs p21/WAF1)

Done

Lines 229- 230: delete ‘… (21-kD protein product of the cyclin-229 dependent kinase inhibitor also known as CDK-interacting protein 1) ...'

Done

Lines 247 and 256: please choose one way of designating the same thing (caspase 9 and caspase 3 vs caspase-3 and caspase-8)

Done

Lines 260-261: instead '...reactive oxygen species...' use abbreviation ROS

Done

Line 266: delete’… as reported in...'

Done

Line 267: correct Estrogen Receptor

Done

Line 268: correct ECGC (EGCG)

Done

Line 308: the line can’t start with a comma

Done

Line 315 correct Genistein

Done

Lines 328-332: C-33A cells are HPV negative

We have deleted Silymarin from the list of compounds since it has been tested only against a HPV-negative cervical cancer cell model.

Line 335: correct bcl2

Done

Line 342: correct Hela

Done

Lines 359-362: C-33A cells are HPV negative so it is not possible that curcumin has '...downregulation of the viral oncogenes E6 and E7...'

We have improved the text to distinguish general anti-cancer effects from HPV-specific effects of curcumin and to better focus on the latter (lines 400-404).

Line 437: correct bBrussels

Done

Lines 437-480: please be consistent with numeration '...CIN II or CIN III...' or '... CIN2....CIN3...' The authors should explain abbreviations when they are first introduced.

Done

Line 497: correct Ddecursinol; Ppyranocoumarin compounds

Done

Line 504: correct Ssteroid

Done

Line 506: correct Wwithaferin

Done

Lines 509-513: I can't notice the connection to HPV

Thanks. We agree and we have deleted the related paragraph (lines 559-563)

Line 546: correct Ppolyphenon E

Done

Line 551: the authors should not start a sentence with a number. Please rewrite.

The number is now preceded by ‘Indeed’.

Line 556: correct Pphase

Done

Line 562: correct Ffucoidan

Done

Lines 566-568: I consider this sentence sufficient and unnecessary ‘The Ru-Fu complex has been demonstrated to be compatible with cells and human RBCs and the release of rutin from the Ru-Fu complex has been shown to occur preferentially at endosome pH 5.5.’

The sentence has been erased.

Line 595: correct Bbryophyllin

Done

Line 603: correct Pphase

Done

Lines 604, 606 and 610: correct Bbasant

Done

Line 637: correct in-silico

Done

Line 647-736: too long a paragraph and out of focus of paper. Please shorten.

The paragraph has been shortened to half and references related to the deleted sections have been consequently erased from bibliography.

Line 656: correct [167,168][169] – [167-169]

Done

Line 677: phyto-compounds without '-'

Done

Lines 745-747: delete ‘Indeed, it has been shown that ι-carrageenans can inhibit HPV infection with significantly higher efficiency than heparin.’

The sentence has been deleted

Lines 760, 775, 801, 814: correct Hela

Done

Line 800: correct Stichopus Japonicus

Done

Line 805: correct anti HPV

Done

Discussion

Line 832: correct Developing Countries

Done

Line 833: correct Developing Countries

Done

Line 896: correct Developing Countries

Done

Line 900: correct Ccountries

Done

Line 901: correct Ddiindolylmethane; abreviation DIM is introduced earlier

Done

Line 903: ‘… acidic environment of the gut...' The authors should check this information because intestines have an alkaline pH.

We have changed ‘gut’ with ‘stomach’ (line 958)

Figure 1 correct Ssubstances

Done

Figure 2 needs to be improved. Plant cells (if this is a plant cell) is not connected with anything.

The plant cell wall now includes all the class of plant molecules tested for anti-HPV properties.

Figure 3 better description is needed; abbreviations should be explained; correct Plant Molecular Farming

Done

Reviewer 4 Report

Massa et al. have summarized the literature investigating the effect of natural bioactives and their anti-HPV properties. The review included both basic science and clinical studies. The review was well structured and included a very interesting summary of this topic. I think that this review makes a great contribution in this field.

Author Response

We deeply thank the Reviewer for the very gratifying comments.

Round 2

Reviewer 1 Report

Thanks to the authors for their corrections and additions, this article can be published in this form 

Reviewer 2 Report

I have no further comments.

This manuscript is a resubmission of an earlier submission. The following is a list of the peer review reports and author responses from that submission.

Round 1

Reviewer 1 Report

Massa et al. present a review on natural bioactives of potential therapeutic interest for human papillomavirus infection. In recent years, the utility of bioactives as potential therapeutic supplement is being investigated for a number of sexually transmitted illnesses. As human papillomavirus infection contributed significantly to the burden of sexually transmitted illnesses, including causing (pre)malignant lesions, novel therapeutic options for this virus should prove useful. This review provides an overview of the information relevant in this regard.

Specific comments:

  1. The authors need to identify the nature of the review as narrative / scoping / systematic. They should provide detailed information on the bibliographic databases that were searched, the search strategies that were used, the methodology followed for the inclusion of literature. They need to mention whether the PRISMA guidelines were followed, and provide the PRISMA checklist if the guidelines were followed.
  2. The abstract is vague and non-informative. The authors could state the intention of the review, i.e. the deficits in the current therapeutic options that bioactives could potentially address, and the main findings from the review more clearly.
  3. The meaning of the phrase in line 35 (Introduction) – “in an environmentally endangered context” is unclear.
  4. An important challenge for the treatment of human papillomavirus is the tackling the capability of the virus for latency and recurrence. The authors should discuss whether any bioactive compound has been studied for the management of viral latency / recurrence or whether bioactives could have any potential role for addressing viral latency / recurrence.
  5. Lines 102 – 107 (Introduction) appear speculative and theoretical. The authors should be more specific about the objectives of this review.
  6. The authors need to discuss how potentially these bioactives could be used for drug development, i.e. the additional research that will be required for the translation of bioactives as prescription medication.
  7. Mechanism of action of all bioactive subclass need to provided consistently throughout the manuscript.
  8. A number of recent studies have argued that usage of curcumin supplements reduce the efficacy of chemotherapeutic agents. For each subgroup of the bioactives presented, the authors should discuss the potential down-sides / reported drug interactions / adverse effects.
  9. Line 311: The authors should present the parameters by means of which the risk reduction of cervical cancer in Chinese women were measured.
  10. The authors should present the information on the level of evidence available for the utility of the bioactives for the treatment of HPV infection, including the model / trial where these were tested, and the IC50 as a Table.

Reviewer 2 Report

This review is very interesting, original and complete.

Nevertheless, it is necessary to explain the methodology of this review, which databases have been consulted, with which keywords and which search equations, a review must be reproducible in its methodology to be completed later.

A solution must be found to present figure 1 differently.

Round 2

Reviewer 1 Report

The authors have largely addressed the comments that were made on the original submission, including adding some interesting visualisations and the IC50s of the bio-actives. The manuscript has considerably improved in terms of quality.

I have some additional comments:

  1. My major concern is that the authors seem to have misplaced some of the references. For example, on page 7, lines 277 – 280, the authors mention that a genome-wide study has reported the capability of EGCG to alter DNA methylation in oral squamous cell carcinoma. First of all, it needs to be mentioned that these results were obtained from analyses performed on oral squamous cell carcinoma cell lines. Simply mentioning oral squamous cell carcinoma seems misleading. Moreover, the reference cited here, i.e., reference number 77, seems to have been incorrectly cited, because the cited paper by Chen et al. reports on protective effect of EGCG against nephrotoxicity induced by cisplatin. In the next lines, the authors describe the role of EGCG in regulating miRNA expression in certain cell lines. They cite reference number 77, which also seems to have been incorrectly cited. Reference number 78, i.e., the study by Zhou et al. reports the immunomodulatory effect of EGCG in Parkinsonism. Zhou et al. studied T-cell subpopulations in this study using flow cytometry analyses and did not study cell lines.

Incorrect citations are also repeated on pages 10 and 18. On page 10, reference number 124, cited in the section on Allicin, is actually a study by Ahn et al. on the inhibitory role of EGCG on human cervical cancer cell lines. On page 18, reference number 205, cited in the section on Algae, is a study by Mahumud RA et al. on the cost-effectiveness of the nonavalent HPV vaccines.

These are major deficiencies of the manuscript and the authors need to address this.

  1. Page 19, lines 826 – 833, the authors describe the role of hemocyanins in the inhibition of oral cancer in mouse models. The authors need to highlight better that human vaccine had also been administered in this study in addition to hemocyanins. In addition, the authors mention “oropharyngeal carcinoma”, whereas the reported study, i.e., reference number 216, had been performed on mouse models of oral (not oropharyngeal) carcinoma.
  2. The authors have added in the methodology section that the review is narrative / scoping in nature. They have provided information on the bibliographic databases that were searched. They should add the complete search strategies that were used in the supplement.
  3. Minor comment: The manuscript still contains a number of grammatical and typographical errors. The authors need to perform a thorough proof-reading of the manuscript.

Author Response

RESPONSES TO THE REVIEWER

We kindly thanks the Reviewer for its precious indications. Revisions are marked in the text and specific responses to the referee comments are reported in the following paragraphs.

Specific comments:

  1. My major concern is that the authors seem to have misplaced some of the references. For example, on page 7, lines 277 – 280, the authors mention that a genome-wide study has reported the capability of EGCG to alter DNA methylation in oral squamous cell carcinoma. First of all, it needs to be mentioned that these results were obtained from analyses performed on oral squamous cell carcinoma cell lines. Simply mentioning oral squamous cell carcinoma seems misleading. Moreover, the reference cited here, i.e., reference number 77, seems to have been incorrectly cited, because the cited paper by Chen et al. reports on protective effect of EGCG against nephrotoxicity induced by cisplatin. In the next lines, the authors describe the role of EGCG in regulating miRNA expression in certain cell lines. They cite reference number 77, which also seems to have been incorrectly cited. Reference number 78, i.e., the study by Zhou et al. reports the immunomodulatory effect of EGCG in Parkinsonism. Zhou et al. studied T-cell subpopulations in this study using flow cytometry analyses and did not study cell lines.

Incorrect citations are also repeated on pages 10 and 18. On page 10, reference number 124, cited in the section on Allicin, is actually a study by Ahn et al. on the inhibitory role of EGCG on human cervical cancer cell lines. On page 18, reference number 205, cited in the section on Algae, is a study by Mahumud RA et al. on the cost-effectiveness of the nonavalent HPV vaccines.

As recommended by the reviewer, the section ‘3.1.2.3 Flavonols’ was rephrased.

As suggested, we corrected the wrong references introduced by homonymy by Mendeley software as follow:

‘Chen, B.; Liu, G.; Zou, P.; Li, X.; Hao, Q.; Jiang, B.; Yang, X.; Hu, Z. Epigallocatechin-3-gallate protects against cisplatin-induced nephrotoxicity by inhibiting endoplasmic reticulum stress-induced apoptosis. Exp. Biol. Med. 2015, 240, 1513–1519, doi:10.1177/1535370215573394.’

has been replaced with

‘Chen, L., Han, W., Geng, Y., & Su, J. (2015). A genome-wide study of DNA methylation modified by epigallocatechin-3-gallate in the CAL-27 cell line. Molecular Medicine Reports, 12, 5886-5890. https://doi.org/10.3892/mmr.2015.4118’.

‘Zhou, T.; Zhu, M.; Liang, Z. (-)-Epigallocatechin-3-gallate modulates peripheral immunity in the MPTP-induced mouse model of Parkinson’s disease. Mol Med Rep 2018, 17, 4883–4888, doi:10.3892/mmr.2018.8470.’

has been replaced with

‘Zhu, Y., Huang, Y., Liu, M., Yan, Q., Zhao, W., Yang, P. ... Ma, L. (2019). Epigallocatechin gallate inhibits cell growth and regulates miRNA expression in cervical carcinoma cell lines infected with different high‑risk human papillomavirus subtypes. Experimental and Therapeutic Medicine, 17, 1742-1748. https://doi.org/10.3892/etm.2018.7131’.

The reference n.124 was not relevant and has been eliminated.

‘Mahumud, Rashidul Alam, et al. "Cost-effectiveness evaluations of the 9-Valent human papillomavirus (HPV) vaccine: Evidence from a systematic review." PloS one 15.6 (2020): e0233499.’

has been replaced with

‘Wang, S.-X.; Zhang, X.-S.; Guan, H.-S.; Wang, W. Potential Anti-HPV and Related Cancer Agents from Marine Resources: An Overview. Mar. Drugs 2014, 12, 2019-2035. https://doi.org/10.3390/md12042019’.

Moreover, a thorough revision of bibliography and corresponding numbering was performed.

These are major deficiencies of the manuscript and the authors need to address this.

  1. Page 19, lines 826 – 833, the authors describe the role of hemocyanins in the inhibition of oral cancer in mouse models. The authors need to highlight better that human vaccine had also been administered in this study in addition to hemocyanins. In addition, the authors mention “oropharyngeal carcinoma”, whereas the reported study, i.e., reference number 216, had been performed on mouse models of oral (not oropharyngeal) carcinoma.

We agree with the Reviewer’s comment that the mouse model used is for oral cancer and therefore we modified the sentence for better clarity. In addition, we described in more detail the adjuvant that was used in the mentioned study as a co-treatment with hemocyanins.

  1. The authors have added in the methodology section that the review is narrative / scoping in nature. They have provided information on the bibliographic databases that were searched. They should add the complete search strategies that were used in the supplement.

As stated in the JCM ‘Istructions for authors’ section, the description of methodology is recommended only in the case of systematic reviews and meta-analyses (Prisma checklist). As mentioned, our review is a narrative review that we wanted to be slightly reinforced by an approach that could be more resembling a scoping methodological framework. According to Reviewer’s request, we added information about search strategies in a supplementary file.

  1. Minor comment: The manuscript still contains a number of grammatical and typographical errors. The authors need to perform a thorough proof-reading of the manuscript.

A thorough proof-reading of the manuscript was done.

Reviewer 2 Report

Thanks to the authors for their addition and correction.

Author Response

We deeply thank the Reviewer.